# Study on Degradation of Natural Rubber Latex Using Hydrogen Peroxide and Sodium Nitrite in the Presence of Formic Acid

**DOI:** 10.3390/polym15041031

**Published:** 2023-02-19

**Authors:** Kraiwut Wisetkhamsai, Weerawat Patthaveekongka, Wanvimon Arayapranee

**Affiliations:** 1Department of Chemical Engineering, Faculty of Engineering and Industrial Technology, Silpakorn University, Muang, Nakorn Pathom 73000, Thailand; 2Department of Chemical Engineering, College of Engineering, Rangsit University, Muang, Pathumthani 12000, Thailand

**Keywords:** liquid natural rubber, degradation, thermal stability

## Abstract

Liquid natural rubber (LNR), a depolymerized natural rubber (NR) consisting of shorter chains, was prepared via oxidative degradation using NaNO_2_ and H_2_O_2_ degrading agents in the presence of HCOOH. The influence of reagent concentrations, temperature, and reaction time on the number-average molecular weight (M_n_) was studied. Results showed the higher concentration of H_2_O_2_ and HCOOH employed faster degradative rates. However, a higher concentration of NaNO_2_ decreased the M_n_ reduction. Prolonged reaction time and high temperature resulted in a product with low M_n_. FTIR spectra indicated the synthesized LNR contained hydroxyl end groups resulting from the breaking of the NR chains at an acidic pH, whereas a carboxyl terminated LNR was formed at an alkaline pH. SEM micrographs showed the latex particles of LNR were spherical and smaller compared to NR. The experimental results showed the reaction orders of [H_2_O_2_], [HCOOH], and [NaNO_2_] were 1.58, 0.79, and −0.65, respectively. In addition, the pre-exponential factor and activation energy were 1.04 × 10^9^ M^−1.72^ t^−1^ and 78.66 kJ/mol, respectively. Based on TGA analysis, the thermal stability of the rubber depended on its M_n_. The LNR containing functional end groups exhibited thermal instability and could be a starting material for many applications.

## 1. Introduction

Natural rubber (NR) is an important renewable biopolymer in many industries due to its elastic property and excellent mechanical strength. NR latex is known as a diene rubber in its repeating units of cis-1,4-isoprene, with the molecular weight varying greatly, from only a few hundred thousand to more than one million [1]. The NR can be degraded to liquid natural rubber (LNR), which has a similar microstructure to NR and consists of shorter polymeric chains and lower molecular weight (M_w_ < 10^5^) [2]. It becomes a modified NR, allowing greater flexibility in the production of the NR used. LNR produced from the degradation of NR latex is related to breaking the NR chain either by direct scission of the carbon–carbon single or carbon–carbon double bonds of the polyisoprene backbone. The molecular weight of NR can be reduced by different methods such as mechanical, chemical, photochemical, ozonolysis, or thermal degradation in a dry, solution, or latex state. Mechanical degradation of NR through mastication in the air at high temperatures is usually done before processing to facilitate easy processing or incorporation of other chemicals [3]. Ibrahim et al. [4] prepared the degraded NR latex under UV light in the presence of H_2_O_2_ and nano TiO_2_, which was prepared via the sol–gel method. The NR chains were attacked by radicals at carbon–carbon single bond or oxidation at carbon–carbon double bond by oxygen species. LNR containing hydroxyl and carbonyl end groups was observed, resulting from chain breaking. Sillapasuwan et al. [5] synthesized the deproteinized LNR with hydroxyl end groups by the photochemical reaction. This reaction was carried out through photocatalytic decomposition of H_2_O_2_ when exposed to UV light, with or without a TiO_2_ film mounted on the quartz substrate. LNR was used as a compatibilizer in NR/LLDPE (linear low-density polyethylene) blends [6,7,8]. Dahlan et al. [6] observed the effects of various molecular weights of LNR as a compatibilizer on the physical properties of NR/LLDPE blends. The ultraviolet irradiation technique was used to prepare LNR. Adding LNR to the blends improved the interaction between the phase of the blends, and they concluded that the LNR content added to the blends marginally affected the physical properties of NR/LLDPE blends. Moreover, LNR with a molecular weight below 20,000 g/mol is an attractive material for enhancement of further modifications. The potential application of LNR with functional end groups is used as a precursor for new material. The authors of [9,10,11] prepared flexible foam from poly(ε-caprolactone) diol (PCL) and hydroxyl telechelic NR (HTNR). First, carbonyl telechelic NR (CTNR) was prepared by an oxidative chain cleavage reaction in tetrahydrofuran with periodic acid. After that, sodium borohydride was then used to change the LNR with carbonyl end group (CTNR) into LNR with hydroxyl end group (HTNR). They claimed that HTNR-based polyurethane foams had improved elastic properties and low-temperature flexibility compared to foam based on commercial polyol. LNR with a carboxyl termination and its subsequent application in NBR compounds as a reactive polymeric plasticizer was described by Dileep et al. [12]. They found that during sulfur vulcanization, the natural rubber component of the carboxyl terminated LNR attacked the NBR. Consequently, this plasticizer was neither volatile nor extractable. Nor and Ebdon [13] studied the use of gel permeation chromatography (GPC) and Fourier transform infrared (FTIR) spectroscopy to follow the kinetics of chain scission and major changes in functional groups of NR during ozonolysis in chloroform at 0 °C. They reported that the introduction of a variety of oxygenated functional groups accompanied the reduction in molecular weight of NR during ozonolysis. Many investigations concentrated on LNR [14,15,16] and deproteinized LNR (LDNR) [17] via oxidatively degraded using H_2_O_2_ and NaNO_2_ reagents. The broken chains had different end groups depending on the pH of the reaction medium. Bac et al. [14] studied the in situ epoxidation of natural rubber latex in chain-scissoring conditions. The epoxide ring underwent a ring-opening process at low pH and prolonged time, forming hydroxyl, carbonyl, and ether groups. Isa et al. [15] investigated the degradation of NR using peroxide acid in the presence of NaNO_2_. They found that NR depolymerized into LNR with an epoxyl and a hydroxyl group. The rate of degradation was affected by reaction temperature and time. Ibrahim et al. [16] proposed the degradation mechanisms were different as H_2_O_2_ and NaNO_2_ underwent different routes in acidic and alkaline media. The end groups of the LNR samples performed in acidic and alkaline media were found to be hydroxyl and carbonyl, respectively. Fadhillah et al. [18] studied LNR production and the effect of CoCl_2_ catalyst and NaNO_2_ levels on the molecular weight of LNR. They concluded that reaction time and NaNO_2_ and CoCl_2_ concentrations affected the molecular weight of the LNR produced. Phetphaisit and Phinyocheep [19] investigated the kinetics of degradation of NR using a combination of potassium persulfate (K_2_S_2_O_8_) as a radical initiator and propanal as a secondary reagent. They reported that the intrinsic viscosity [η] of the LNR obtained depended on the initiator concentration, amount of propanol, dry rubber content, reaction time, and temperature. The degradation rate exhibited a second-order rate law dependence on the cleavage of a single carbon–carbon bond of a polyisoprene chain. The activation energy of the degradation reaction was 76.56 kJ/mol.

Many relevant investigations, including the effects of various chemicals and environmental conditions as well as the mechanism of NR degradation in acidic and basic environments, have been previously carried out. However, rarely have studies on the rate of NR degradation been reported. The objectives of this research were to prepare liquid natural rubber with oxidative degradation by using hydrogen peroxide (H_2_O_2_), formic acid (HCOOH), and sodium nitrite (NaNO_2_) as reagents. Intrinsic viscosity ([η]) was characterized by the dilution method using an Ubbelohde capillary viscometer at 30 °C when toluene has operated as a solvent. The number-average molecular weight of LNR was calculated by using Mark–Houwink equation [20]. The efficiency of the degradation of NR was determined by measuring its number-average molecular weight (M_n_). The effects of reaction parameters such as H_2_O_2_, HCOOH, and NaNO_2_ concentrations, reaction temperature, and reaction time were investigated. The LNR was then characterized for its chemical structure using FTIR, while scanning electron microscopy (SEM) was used to observe the morphology of the latex particles. The degradative kinetics of the NR to form LNR based on molecular weight data have been evaluated. The thermal decomposition of the rubber samples was studied in terms of thermogravimetric analysis (TGA) and derivative thermogravimetric (DTG).

## 2. Materials and Methods

### 2.1. Materials

High ammonia natural rubber latex (60% dry rubber content) used in the present study was purchased from Bothong Natural Rubber Trade Co., Ltd., Bothong, Chonburi, Thailand. 50 wt% aqueous hydrogen peroxide (H_2_O_2_), 85 wt% formic acid (HCOOH), 97 wt% sodium nitrite (NaNO_2_), and 98 wt% sodium sulfite (Na_2_SO_3_) were purchased from Sigma-Aldrich, Missouri, USA. Polyoxyethylene styrenated phenyl ether (C_24_H_22_O_2_) used as a surfactant under the trade name Emulvin WA was purchased from Chemical and Materials Co., Ltd., Map Pong, Chom Thong, Bangkok, Thailand. Methanol (commercial grade) was bought from Facobis Co., Ltd. (Bangkok, Thailand). All chemical reagents were used as received. Distilled water was used throughout the work.

### 2.2. Degradation of Natural Rubber Latex

A total of 227 g of the NR latex, having 60% DRC, was mixed with Emulvin WA and distilled water in a flask equipped with a condenser and stirrer. The NR latex was acidified by formic acid to the required pH followed by successive treatment with 50% aqueous H_2_O_2_ and 10% solution of NaNO_2_. The reaction mixture was diluted to 1 L with distilled water and kept in latex form at a constant temperature for desired time. After the reaction, 5% sodium sulfite solution was added to the mixture to remove excess H_2_O_2_ [21], followed by precipitation with methanol. Then precipitated LNR was washed two times with distilled water and dried in a vacuum oven at 40 °C until a constant weight was achieved.

### 2.3. Characterization

#### 2.3.1. Number-Average Molecular Weights

The rubber was further purified by dissolving it in toluene at room temperature for 24 h. After the sample had been dissolved, the solution was transferred by passing through a cotton filter and re-coagulated into methanol. The coagulated rubber product was washed three times with distilled water and then dried to a constant weight in a vacuum oven at 40 °C.

Viscometry, a popular method used to determine molecular weight, is quick and simple and requires a relatively inexpensive apparatus. According to ASTM D445, the Ubbelohde viscometer was used to determine the intrinsic viscosities of the rubber samples in toluene at 30 °C by extrapolation to zero concentrations of specific viscosity measurements at five different concentrations.

The average molecular weights of the rubber samples were determined by gel permeation chromatography (GPC, Alliance, Waters e2695 separations, Milford, MA, USA.) using tetrahydrofuran (THF) as eluent and polystyrene for standard calibration at 30 °C. The rubber sample was dissolved in THF and filtered using a 0.45 μm syringe filter before injecting it into a chromatograph.

Based on the Mark–Houwink equation, the relationship between the intrinsic viscosity and average molecular weight was built for the rubber sample in toluene at 30 °C. The slice log molecular weight curve of LNR samples with different intrinsic viscosities is shown in Figure 1. The relationship between number-average molecular weight (M_n_) and [η] can be described as Equation (1):(1)$$[\eta ]=KM_{n}^{a}$$
where K and a are constants for a certain solvent and temperature. Figure 2 shows the plot of log [η] versus log M_n_ for rubber samples, which yielded a straight line, and the values of a and K were obtained from the slope and the intercept of this line, respectively. These K and a values of the Mark–Houwink equation were further used to transform the [η] of the rubber sample into M_n_. Then the following Equation (2) was obtained:(2)$$[\eta ]=1.307\;\times \;{10^{-}}^{3}M_{n}^{0.638}$$

#### 2.3.2. FTIR Spectroscopy

The chemical structure of rubber samples was determined by attenuated total reflectance-Fourier-transform infrared (ATR-FTIR) spectroscopy (Perkin Elmer Frontier, CT, USA.). The samples were analyzed in transmittance mode within the range 600–4000 cm^−1^.

#### 2.3.3. Morphology

The morphology of rubber particles was analyzed using a field emission-scanning electron microscope (FE-SEM, Tescan, Mira3, Brno, Czech Republic.). Samples were coated with the thin gold layer and detected by backscattered electrons with an acceleration of 5 kV.

#### 2.3.4. Thermal Analysis

The thermal degradation of rubber samples was carried out by a thermogravimetric analyzer (TGA, Perkin Elmer, Pyris 1, CT, USA.). The sample was weighted to 5.0 ± 0.2 mg in alumina crucibles. TGA was conducted from ambient to 800 °C at a heating rate of 40 °C/min under an air flow at 20 mL/min. Continuous recordings of the rubber sample’s temperature, mass, and heat flow were performed.

## 3. Results and Discussion

### 3.1. FTIR Spectroscopy

Infrared spectra of the NR starting material and the obtained LNRs at various pH were compared, as shown in Figure 3. The obtained LNR exhibited the absorption peaks of C=C stretching and =C-H bending at 1665 and 834 cm^−1^, similar to those observed in the NR. The FTIR spectra of LNR samples prepared at different pH showed a few new peaks compared to the NR. For the LNR sample prepared in an acid medium, a broad peak located at 3400 cm^−1^ was assigned for the stretching mode of the hydroxyl groups of LNR chains. Meanwhile, the presence of other groups on the LNR chains prepared in neutral and alkaline media can be attributed to the appearance of carbonyl peaks (C=O) at 1720 cm^−1^ for ketone and 1739 cm^−1^ for aldehyde, respectively.

The chain breaking of NR at the carbon–carbon single bond is usually due to an attack of radicals and leaving a hydroxyl end group. In contrast, an oxidizing agent can oxidize the broken carbon–carbon double bond in the NR chains to leave a carbonyl end group. Mechanisms of LNR samples prepared via oxidative degradation using NaNO_2_ and H_2_O_2_ reagents were found to have different end groups, hydroxyl, and carbonyl, depending on the pH of the system used. H_2_O_2_ and NaNO_2_ reacted to form peroxynitrous acid, which decomposed to form ^•^OH radicals at pH < 7. The hydroxyl radicals attacked the carbon–carbon single bond of polyisoprene chains. Breakage of polyisoprene chains by hydroxyl radicals then yielded LNR with hydroxyl end groups, as shown in Figure 1. At pH ≥ 7, hydrogen peroxide preferred to oxidize carbon–carbon double bond to form epoxy rings, which were attacked by nitrite ion (NO_2_^−^) during degradation, thus yielding chain breaking. The result was that the polyisoprene chains were getting shorter and had carbonyl end groups (Figure 2). Similar mechanisms of LNR with different end groups prepared in different pH conditions were proposed by Ibrahim et al. [16].

### 3.2. Effect of Concentration of Hydrogen Peroxide

The effect of the concentration of H_2_O_2_ on the number-average molecular weight at various reaction times was studied over the range of approximately 0.1 to 0.4 M while keeping the pH, [NaNO_2_], and temperature constant, as shown in Figure 4. At the same time, the M_n_ of LNR was seen to decrease with an increase in the concentration of H_2_O_2_ from 0.1 to 0.4 M. The molecular weight of NR decreased from 630,439 g/mol to 12,325 g/mol, indicating that the NR degraded. This trend can be explained by the fact that a higher concentration of H_2_O_2_ provided more sources of hydroxyl radicals in the reaction, hence enhancing the chain scission of NR (Figure 1). So, a greater concentration of H_2_O_2_ can lead to a faster rate of oxidative degradation of the NR.
Condition: pH = 5, NaNO_2_ = 0.02 M, and T = 338 K.

### 3.3. Effect of Concentration of Formic Acid

The amount of formic acid used was adjusted to the required pH (pH 5–11). We studied the influence of HCOOH concentration on the M_n_ at various reaction times over the range of approximately 0.0187 M (pH 11) to 0.1247 M (pH 5) while keeping the [H_2_O_2_], [NaNO_2_], and temperature constant, as shown in Figure 5. A higher concentration of HCOOH provided more sources of hydrogen ions, hence increasing the amount of peroxynitrite acid releasing ^•^OH and ^•^NO_2_ radicals. Consequently, an increase in HCOOH concentration also increased the rate of degradation of LNR. Based on Figure 5, the degradation of NR at an acidic pH showed to be more effective than at an alkaline pH, thus yielding LNR with lower M_n_.
Condition: H_2_O_2_ = 0.4 M, NaNO_2_ = 0.02 M, and T = 338 K. 

### 3.4. Effect of Concentration of Sodium Nitrile

The effect of the concentration of NaNO_2_ on the number-average molecular weight at various reaction times was studied over the range of approximately 0.02 to 0.2 M while keeping the [H_2_O_2_], pH, and temperature constant, as shown in Figure 6. The results showed that an increasing concentration of NaNO_2_ increased the M_n_ of LNR at the same reaction time. The increase in M_n_ was due to the higher concentration of nitrite ions, which can be oxidized by the ^•^OH radicals, providing an excessive amount of ^•^NO_2_ radicals and hydroxide (OH^−^). The ^•^NO_2_ radicals became exceedingly unstable in an aqueous solution [22,23], so it coupled with itself to form N_2_O_4_, and hydrolysis of N_2_O_4_ in water generated HNO_2_. The HNO_2_ reacted with ^•^OH to produce ^•^NO_2_. The formation of side reactions led to the reduction of ^•^OH radicals that inhibited degradative reaction or triggered crosslinking reaction at higher NaNO_2_ concentrations [24].
Condition: H_2_O_2_ = 0.4 M, pH = 5, and T = 338 K. 

### 3.5. Effect of Reaction Temperature

The effect of the temperature degradation on molecular weight was performed at 328, 333, 338, and 343 K for various reaction times by fixing the concentration of other reagents, as shown in Figure 7. The temperature influenced the efficiency of the degradation reaction. Higher reaction temperatures provided more energy to excite the molecules and allowed many chain-scissoring reactions to occur faster. Thus, an increase in reaction temperature also increased the rate of degradation of NR.
Condition: H_2_O_2_ = 0.4 M, pH = 5, NaNO_2_ = 0.02 M.

### 3.6. Effect of Reaction Time

The effect of reaction time on M_n_ of LNR was investigated by varying the reaction time from 3 to 12 h. It can be noticed that the M_n_ values of all rubber samples were exponentially reduced according to the increased reaction time, as shown in Figure 4, Figure 5, Figure 6 and Figure 7. With an increase in reaction time, the M_n_ values in all cases tended to decrease significantly during the first 3 h and then decreased after 3 h of reaction time. There was a gradual decrease in the M_n_ when the reaction time was extended to 9 and 12 h. The morphology of NR and LNR particles at different molecular weights was investigated with an SEM micrograph, as shown in Figure 8a–d. There was much difference in the case of latex particle morphology. The spherical and pear shape had the biggest sizes before the oxidative degradation, as shown in Figure 8a. According to the above-mentioned factors, prolonged reaction times allowed more degradation of rubber chains into smaller particles. It was found that some of the latex particles were spherical, smaller, and had a wider size range compared to the NR particles after 3 h of degradative reaction (Figure 8b). After 12 h of degradation, Figure 8d shows that the latex particles were almost uniform and the smallest in size.
Condition: H_2_O_2_ = 0.4 M, pH = 5, NaNO_2_ = 0.02 M, and T = 338 K.

### 3.7. Degradative Kinetics

The degradative reaction of the LNR chains can be determined as a second-order reaction with respect to the number of bonds broken during depolymerization [25,26,27]. Referring to Equation (2), the intrinsic viscosity obtained can be calculated to the M_n_ values. A chain scission model for random scission of the macromolecule can be derived as in Equation (3):(3)$$\frac{1}{M_{n(t)}}\;-\;\frac{1}{M_{n(0)}}=\frac{\mathrm{kt}}{M_{0}}$$
where M_n(t)_ is the M_n_ of LNR at time = t, M_n(0)_ is the M_n_ of NR at time = 0, M_0_ is the M_n_ of isoprene unit (68 g/mol), and k is the rate constant. As previously discussed, process parameters, i.e., H_2_O_2_ concentration, HCOOH concentration, NaNO_2_ concentration, and temperature, significantly influence the degradation process. According to the Arrhenius relation, k = A×Exp(−E_a_/RT) where A, E_a_, R, and T are the pre-exponential factor, activation energy, gas constant, and temperature, respectively. These parameters have been proposed to relate to the kinetic model as a power function. Therefore, Equation (3) can be rewritten as Equation (4):(4)$$\frac{M_{0}}{M_{n(t)}}\;-\;\frac{M_{0}}{M_{n(0)}}=A\times e^{\frac{{-E}_{a}}{\mathrm{RT}}}{{[H}_{2}O_{2}]}^{m}{{[\mathrm{CH}}_{2}O_{2}]}^{n}{{[\mathrm{NaNO}}_{2}]}^{o}\times \;t$$
where [H_2_O_2_], [HCOOH], and [NaNO_2_] are the initial concentration of reagents (mol/L). The exponents of the reactant concentrations m, n, and o are referred to as partial orders of the reaction that must be determined experimentally. The partial order corresponding to each reactant is now calculated by conducting the reaction with varying concentrations of the reactant in question, and the concentrations of the other reactants are kept constant.

Generally, the kinetic model, a power function of the process variables, is independent of time. Referring to Figure 4, Figure 5 and Figure 6, the partial orders can be determined by using a log [(M_0_/M_n(t)_ − M_0_/M_n(0)_)/t]-log [reactant] plots and the linear regression technique, as shown in Figure 9, Figure 10 and Figure 11. The corresponding slopes were the partial orders given by m, n, and o, respectively.
Condition: pH = 5, NaNO_2_ = 0.02 M, and T = 338 K. 
Condition: H_2_O_2_ = 0.4 M, NaNO_2_ = 0.02 M, and T = 338 K.
Condition: H_2_O_2_ = 0.4 M, pH = 5, and T = 338 K.

Referring to Figure 7, a plot of ln [(M_0_/M_n(t)_ − M_0_/M_n(0)_)/([0.4]^1.58^[0.1247]^0.79^[0.02]^−0.65^×t)] versus 1000/T gave a straight line whose slope is equal to −1000E_a_/R and whose intercept with the *y* axis is ln A, as shown in Figure 12.
Condition: H_2_O_2_ = 0.4 M, pH = 5, and NaNO_2_ = 0.02 M.

The reaction orders with respect to H_2_O_2_ concentration and HCOOH concentration deviated from 1, indicating a nonlinear relationship between the degradative reaction. An increase in reaction temperature and the concentrations of H_2_O_2_ and HCOOH enhanced the chain-scissoring reaction. On the other hand, the negative power of NaNO_2_ concentration indicated that the M_n_ reduction was inversely proportional to NaNO_2_ concentration. Hence, an increase in NaNO_2_ concentration retarded the depolymerization rate. The reaction orders of [H_2_O_2_], [HCOOH], and [NaNO_2_] were 1.58, 0.79, and −0.65, respectively (Figure 9, Figure 10 and Figure 11). According to Figure 12, The pre-exponential factor and activation energy were found to be 1.04 × 10^9^ M^−1.72^ t^−1^ and 78.66 kJ/mol, respectively, according to Figure 12. As a result, the kinetic model for random scission of NR using H_2_O_2_, HCOOH, and NaNO_2_ as reagents can be summarized as Equation (5):(5)$$\frac{M_{0}}{M_{n(t)}}-\frac{M_{0}}{M_{n(0)}}=1.04\times 10^{9}{\;M}^{-1.72}{\;t}^{-1}\;e^{\frac{-78.66\;\cdot \mathrm{kJ}\cdot {\mathrm{mol}}^{-1}}{\mathrm{RT}}}{{[H}_{2}O_{2}]}^{1.58}{{[\mathrm{CH}}_{2}O_{2}]}^{0.79}{{[\mathrm{NaNO}}_{2}]}^{-0.65}\;\times t$$

Figure 13 shows a satisfactory relationship between the experimental and predicted values obtained from Equation (5) of degradation of NR. The kinetic parameters were estimated based on experimental data. When R^2^ was equal to 0.9771, the data prediction showed good performance and a high correlation between the predicted value obtained from Equation (5) and the experimental value.

### 3.8. TGA Analysis

The curves of NR and LNR samples in the air atmosphere were investigated by a thermogravimetric analysis (TGA) instrument, which described the decomposition behavior of NR and LNR at different number-average molecular weights, as shown in Figure 14. Referring to Figure 14a, thermogravimetric (TG) curves are weight loss against temperature at a heating rate of 40 °C/min. The decomposition of NR and LNR samples occurred one step in the temperature range of 50–850 °C. The thermal degradation temperatures of NR and LNR at different molecular weights are listed in Table 1. The initial decomposition temperature (T_i_) and final decomposition temperature (T_f_) using a bitangent method were obtained TGA curve, and temperature at maximum weight loss rate (T_m_), which can be obtained from the peak of the DTG curves, as shown in Figure 14b. It can be seen that the thermal degradation temperature of LNR samples decreased with a decrease in the M_n_. NR had a T_i_, T_m_, and T_f_ of 392 °C, 429 °C, and 469 °C, respectively, while all LNRs exhibited lower degradation temperatures when compared with NR. The T_m_ of LNR decreased from 428 °C to 398 °C when the M_n_ decreased from 50,023 to 10,825 g/mol. The thermal degradation curves of LNR, compared to NR, shifted to lower temperatures. This shows that the reduction in molecular weight led to thermal instability resulting in a reactive material, which can be more prone to reacting with lower energy usage.

## 4. Conclusions

Natural rubber (NR), which has a molecular weight of a few hundred thousand to more than one million, is found in the form of latex. The natural rubber depolymerization process used NaNO_2_ and H_2_O_2_ degrading agents in the presence of HCOOH to produce liquid natural rubber (LNR) with a relatively low molecular weight. This study resulted in the Mark–Houwink equation for the value of the constant K, and a for NR and LNR samples were 1.307 × 10^−3^ and 0.638 respectively, which were used to determine the number-average molecular weight. The molecular weight of LNR obtained depended on concentrations of H_2_O_2_, HCOOH, and NaNO_2_, temperature, and reaction time. It was found that the higher concentration of H_2_O_2_ and HCOOH employed faster rates of degradation. However, a higher concentration of NaNO_2_ decreased the M_n_ reduction. The high degradative temperature made the NR chains break more rapidly. The M_n_ values were a rapid initial decrease followed by a more gradual fall according to the increased reaction time. The results showed that the lowest M_n_ of the resulting LNR product was 10,825 g/mol from the initial molecular weight of 630,439 g/mol. Based on FTIR analysis, the presence of hydroxyl terminal groups resulting from the breaking of the NR chains was synthesized at acidic pH. In contrast, LNR with carbonyl terminal groups was formed as the degradative reaction at alkaline pH. SEM micrographs showed the NR latex particles were small spheres with a few larger pear shape particles. After the degradation process, the LNR latex obtained showed distinctly smaller spherical-shaped particles as compared to the NR particles. Changes in the molecular weight were used to construct a kinetic model for the depolymerization process: (M_0_/M_n(t)_) − (M_0_/M_n(0)_) = [(1.04 × 10^9^ M^−1.72^t^−1^Exp(78.66 kJ mol^−1^)][H_2_O_2_]^1.58^[HCOOH]^0.79^[NaNO_2_]^−0.65^ t. The kinetic model is useful to estimate the number-average molecular weight for degraded NR under different environmental conditions such as H_2_O_2_, HCOOH, and NaNO_2_ concentrations, reaction temperature, and reaction time. From TGA measurements, the decomposition temperatures decreased with a decrease in the M_n_, indicating that the LNR had worse thermal stability than NR. Therefore, LNR should be widely used as a starting material for synthesizing biopolymers.

## Data Availability

All the data is included in the article.

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
