# Peer review of "Study on Degradation of Natural Rubber Latex Using Hydrogen Peroxide and Sodium Nitrite in the Presence of Formic Acid"

_polymers, 2023, doi:10.3390/polym15041031_

Round 1
Reviewer 1 Report
Natural rubber is widely used in many industries, which is the main raw material of rubber products. Degradation behavior of NR is important for material performance. Authors studied the influence of reagent concentrations, temperature, and reaction time on the number-average molecular weight. The current form of this study cannot be acceptable. Some aspects as listed below:
1. There are some grammatical errors in the manuscript. it is recommended to check the full paper.
In Page 3, “The popular method is the viscometer method used to determine the other average molecular weight due to quick and simple.”
In Page6, “Ibrahim et al. [20] reported the chain length of NR was reduced using H2O2 and NaNO2 as reagents in latex form, hence leaving different end groups such as hydroxyl or carbonyl depending on the different pH of the reaction media.”
……
2. More details about the degradation can be given.
3. The influence of NaNO2 and H2O2 degrading agents with the presence of HCOOH have been studied. However, it is suggested to systematically analyze its mechanism of the degradation.
4. Figure 14 should be improved.
5. The value and novelty of the study need to be further refined.
6. Few references in the past five years are present here.
Author Response
Author's Response to Decision Letter for
Study on Degradation of Natural Rubber Latex using Hydrogen Peroxide and Sodium Nitrite in The Presence of Formic Acid (polymers-2178005)
Dear Reviewer,
We thank the Reviewer for taking the time and effort necessary to improve the manuscript's quality. Our responses to the Reviewer’s comments are described below:
Response to Reviewer 1 Comments
|
Natural rubber is widely used in many industries, which is the main raw material of rubber products. Degradation behavior of NR is important for material performance. Authors studied the influence of reagent concentrations, temperature, and reaction time on the number-average molecular weight. The current form of this study cannot be acceptable. Some aspects as listed below.
|
|
|
1. |
There are some grammatical errors in the manuscript. it is recommended to check the full paper. Response: we checked and corrected the grammatical errors in the manuscript carefully.
|
|
|
In Page 3, “The popular method is the viscometer method used to determine the other average molecular weight due to quick and simple.” Response: “Viscometry, a popular method used to determine molecular weight, is quick and simple and requires a relatively inexpensive apparatus.” was replaced. (Line 140-141)
|
|
|
In Page 6, “Ibrahim et al. [20] reported the chain length of NR was reduced using H2O2 and NaNO2 as reagents in latex form. Hence having different end group such as hydroxyl or carbonyl depending on the different pH of the reaction media.” Response: “The chain breaking of NR at carbon-carbon single bond is usually due to an attack of radicals and leaving a hydroxyl end group. In contrast, an oxidizing agent can oxidize the broken carbon-carbon double bond in the NR chains to leave a carbonyl end group.” was replaced. (Line 202-204)
|
|
2. |
More details about the degradation can be given. Response: “LNR produced from the degradation of NR latex is related to breaking the NR chain either by direct scission of the carbon-carbon single or carbon-carbon double bonds of the polyisoprene backbone.” was added. (Line 34-37)
|
|
3. |
The influence of NaNO2 and H2O2 degrading agents with the presence of HCOOH have been studied. However, it is suggested to systematically analyze its mechanism of the degradation. Response: “Mechanisms of LNR samples prepared via oxidative degradation using NaNO2 and H2O2 reagents were found to have different end groups, hydroxyl, and carbonyl, depending on the pH of the system used. H2O2 and NaNO2 reacted to form peroxynitrous acid, which decomposed to form •OH radicals at pH < 7. The hydroxyl radicals attacked the carbon-carbon single bond of polyisoprene chains. Breakage of polyisoprene chains by hydroxyl radicals then yielded LNR with hydroxyl end groups, as shown in Scheme 1. At pH ³ 7, hydrogen peroxide preferred to oxidize carbon-carbon double bond to form epoxy rings, which were attacked by nitrite (NO2-) during degradation, thus yielding chain breaking. The result was that the polyisoprene chains were getting shorter and had carbonyl end groups (Scheme 2). Similar mechanisms of LNR with different end groups prepared in different pH conditions were proposed by Ibrahim et al. [16].” was replaced. (Line 204-215)
|
|
4. |
Figure 14 should be improved. Response: New Figure 14 was replaced. (Line 406-407)
|
|
5. |
The value and novelty of the study need to be further refined. Response: “The kinetic model is useful to estimate the number-average molecular weight for degraded NR under different environmental conditions such as H2O2, HCOOH, and NaNO2 concentrations, reaction temperature, and reaction time.” was added. (Line 432-435)
|
|
6. |
Few references in the past five years are present here. Response: New references were added 5 Sillapasuwan, A.; Saekhow, P.; Rojruthai, P.; Sakdapipanich, J. The Preparation of Hydroxyl-Terminated Deproteinized Natural Rubber Latex by Photochemical Reaction Utilizing Nanometric TiO2 Depositing on Quartz Substrate as a Photocatalyst. Polymers. 2022, 14, 2877 - 2889. (Line 45-48)
17 Thuong, N.T.; Manh, N.D.; Tue, N.N. Characterization of liquid deproteinized natural rubber prepared via oxidative degradation. Vietnam J. Chem. 2020, 58, 826-831. (Line 73)
18 Fadhillah, I.; Wiranata, A.; Zahrina, I. Molecular Weight of Liquid Natural Rubber (LNR) Product from the Chemical Depolymerization Process of High Molecular Weight Natural Rubber Latex. J. Phys.: Conf. Ser. 2020, 1655, 012091. (Line 83-86) Please see the attachment (revised manuscript).
|

Reviewer 2 Report
Dear Authors,
The present paper investigates the degradation behavior of natural rubber to smaller fragments influencing different chemicals. Authors have well written the paper, however, some additional corrections to be highlighted before publication as follows
1. What are the differences between present work and the referenced works [3, 15-17]? Innovation in this research should be indicated.
2. It is better to represent scope and objective of this work in a separate paragraph.
3. Could you please explain how NaNO2 retards the degradation in a little more detail?
4. Which atmosphere is used in TGA analysis?
Author Response
Author's Response to the Decision Letter for
Study on Degradation of Natural Rubber Latex using Hydrogen Peroxide and Sodium Nitrite in The Presence of Formic Acid (polymers-2178005)
Dear Reviewer,
We thank the Reviewer for taking the time and effort necessary to improve the manuscript's quality. Our responses to the Reviewer’s comments are described below:
Response to Reviewer 2 Comments
|
Dear Authors, The present paper investigates the degradation behavior of natural rubber to smaller fragments influencing different chemicals. Authors have well written the paper, however, some additional corrections to be highlighted before publication as follows.
|
|
|
1. |
What are the differences between present work and the referenced works [3, 15-17]? Response: “Many relevant investigations, including the effects of various chemicals and environmental conditions as well as the mechanism of NR degradation in acidic and basic environments, have been previously carried out. However, rare works on the rate of NR degradation have been reported.” was added. (Line 94-97)
Innovation in this research should be indicated. Response: “The kinetic model is useful to estimate the number-average molecular weight for degraded NR under different environmental conditions such as H2O2, HCOOH, and NaNO2 concentrations, reaction temperature, and reaction time.” was added. (Line 432-435)
|
|
2. |
It is better to represent scope and objective of this work in a separate paragraph. Response: The objective of this work was separated. (Line 94)
|
|
3. |
Could you please explain how NaNO2 retards the degradation in a little more detail? Response: “The increase of Mn was due to the higher concentration of nitrite ions, which can be oxidized by the •OH radicals, providing an excessive amount of •NO2 radicals and hydroxide (OH-). The •NO2 radicals became exceedingly unstable in an aqueous solution [22-23], so it coupled with itself to form N2O4, and hydrolysis of N2O4 in water generated HNO2. The HNO2 reacted with •OH to produce •NO2. The formation of side reactions led to the reduction of •OH radicals that inhibited degradative reaction or triggered crosslinking reaction at higher NaNO2 concentrations [24].” was replaced. (Line 255-261)
|
|
4. |
Which atmosphere is used in TGA analysis? Response: in the air atmosphere. (Line 380) Please see the attachment (Revised manuscript)
|

Round 2
Reviewer 1 Report
Natural rubber is widely used in many industries, which is the main raw material of rubber products. Degradation behavior of NR is important for material performance. Authors have revised the manuscript according to the comments. The current form of this study can be acceptable after minor revision. Some aspects as listed below:
1. In Figure 4, more details about the kinetic model can be given, here, what is the other parameters?
2. Figure 1 should be improved. The different intrinsic viscosities can be present in figure.
Author Response
Author's Response to Decision Letter
Study on Degradation of Natural Rubber Latex using Hydrogen Peroxide and Sodium Nitrite in The Presence of Formic Acid (polymers-2178005)
Dear Reviewer,
We would like to thank Reviewer for taking the necessary time and effort to review the manuscript. We sincerely appreciate all your valuable comments and suggestions, which helped us in improving the quality of the manuscript. Our responses to the Reviewer’s comments are described below:
Reviewer 1 (Round 2)
|
Natural rubber is widely used in many industries, which is the main raw material of rubber products. Degradation behavior of NR is important for material performance. Authors have revised the manuscript according to the comments. The current form of this study can be acceptable after minor revision. Some aspects as listed below:
|
|
|
1. |
In Figure 4, more details about the kinetic model can be given, here, what is the other parameters? Answer: 1. In Figure 4 “the other parameters were kept constant” was replaced with “keeping the pH, [NaNO2], and temperature constant”. 2. In Figure 5 “the other parameters were kept constant” was replaced with “keeping the [H2O2], [NaNO2], and temperature constant”. 3. In Figure 6 “keeping the concentration of all other reagents and temperature constant” was replaced with “keeping the [H2O2], pH, and temperature constant”.
|
|
2. |
Figure 1 should be improved. The different intrinsic viscosities can be present in figure. Answer: New Figure 1 was replaced. |
